# Weighted-Rank Contrastive Regression for Robust Learning on Imbalance Social Media Popularity Prediction

## Abstract

Social Media Popularity Prediction (SMPP) is the task of forecasting the level of engagement a social media post will receive. It is crucial for understanding audience engagement and enabling targeted marketing strategies. However, the inherent imbalance in real-world social media data, where certain popularity levels are underrepresented, poses a significant challenge. In this study, we leveraged the recent success of contrastive learning and its integration into regression tasks by introducing a Weighted-Rank CR loss to address the data imbalance challenges. Experiments on the Social Media Prediction Dataset demonstrated that our method outperformed the vanilla approach and the current state-of-the-art contrastive regression approach Rank-N-Contrast (Zha et al., 2024).

## 1 Introduction

Social media platforms have become deeply integrated into our daily lives, influencing how we communicate, access information, and consume content. For businesses and brands, social media represents a vast landscape of potential customers and a powerful tool for advertising and engagement. A crucial aspect of influencer marketing is Social Media Popularity Prediction (SMPP), which is the task of forecasting the level of engagement a social media post will receive. This prediction offers invaluable insights for content creators and businesses, guiding content strategies and marketing decisions

A significant challenge in SMPP is the inherent data imbalance. Popularity metrics, such as likes, often exhibit a skewed distribution, with a few posts becoming viral and most receiving mid-to-low engagement. This imbalance hinders traditional machine learning models' ability to accurately predict popularity across the entire spectrum, as some parts of the spectrum may lack sufficient data for effective model training.

While traditional approaches for handling imbalanced data primarily concentrate on categorical targets (He and Ma, 2013; Chawla et al., 2002; Yen and Lee, 2006), many real-world applications involve continuous target variables, often with skewed distributions. For instance, in computer vision, predicting age from facial images involves a continuous target variable that exhibits inherent imbalances. Similar challenges arise in medical applications where health metrics like heart rate and blood pressure, being continuous variables, frequently display skewed distributions across patients.

Yang et al. (2021) identified these challenges as Deep Imbalanced Regression (DIR) and proposed a smoothing approach to harmonize feature and label space distributions, facilitating robust representation learning. Zha et al. (2024) subsequently refined this approach by formulating the ranking loss as a contrastive regression (CR) loss, thereby enhancing feature-label alignment and mitigating the adverse effects of data imbalance.

Building upon Zha et al. (2024)'s work on applying contrastive learning to feature-label alignment, we introduce Weighted-Rank CR loss as a regularizer to further mitigate the data imbalance problem in social media popularity prediction. Our experiment results demonstrate that by incorporating a weighted mechanism into the state-of-the-art model, we can further enhance popularity prediction accuracy. Subsequently, we propose a straightforward end-to-end contrastive regression learning

framework for multi-modal representation learning, a framework that can be readily adapted to more complex architectures.

## 2 LITERATURE REVIEW

### 2.1 SOCIAL MEDIA POPULARITY PREDICTION

Previous approaches to SMPP have employed two primary methods for feature extraction: manually preprocessed features and the utilization of pre-trained models.

**Manually Processed Features**: Earlier studies in social media popularity prediction mostly relied on manually processed features. Jin et al. (2010) employed upload frequency, upload time, and tags to predict image popularity on Flickr. McParlane et al. (2014) incorporated features from visual context (device type, size, orientation), visual content (scene type, number of faces, dominant color), user profile (gender, account type, number of uploads), and tags represented using TF-IDF vectors. Gelli et al. (2015) employed Name-Entity Recognition (NER) on image descriptions, identifying and counting entities like Location, Organization, and Person. These manually processed features have proven valuable and continue to be widely adopted in recent approaches. Ding et al. (2019) and Lai et al. (2020) also incorporated text features like caption length and tag length. While providing valuable insights, these manually processed features required domain expertise and need to be carefully chosen to avoid bias.

**Pre-trained Models**: In recent years, pre-trained deep learning models have emerged as powerful tools for automatically extracting features from multimodal data such as text and images. Notably, Ding et al. (2019) and Xu et al. (2020) employed a ResNet backbone pre-trained on ImageNet for visual features and Word2Vec for textual features. Alternatively, Wu et al. (2022) utilized BERT (Bidirectional Encoder Representations from Transformers) for text and CLIP (Contrastive Language-Image Pre-training) for joint text-image features. These approaches effectively capture intricate patterns that are challenging for manual feature engineering. Figure 1 illustrates a multimodal post encoder proposed by (Kim et al., 2020), which effectively summarizes the feature extraction process for social media posts. The integration of extracted pre-trained multimodal features allows SMPP models to attain enhanced accuracy and robustness, leading to their widespread adoption in recent approaches.

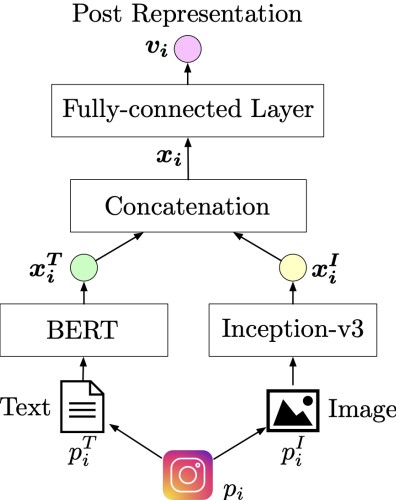

Figure 1: A typical feature extraction framework of social media posts (Kim et al., 2020).

### 2.2 OVERCOME THE IMBALANCE REGRESSION

Despite significant progress in SMPP, a critical challenge remains largely unaddressed: the inherent imbalance within social media data. The popularity distributions of real-world social media data

often exhibits a long-tail pattern, with a small portion of posts having high popularity, while a large number of posts having mid-to-low popularity.

**Re-sampling and Re-weighting**: Traditional re-sampling methods primarily target classification tasks. However, some adaptations have been tailored for imbalanced regression. Random under-sampling (Torgo et al., 2013; 2015) grouped lables in bins and randomly removes samples from majority bins to balance with minority bins. SMOTER (Torgo et al., 2013), a regression adaptation of SMOTE (Chawla et al., 2002), combines undersampling with synthetic minority sample generation to balance the data distribution. SMOGN (Branco et al., 2017) further improves SMOTER by adding gaussian noise to increase sample diversity. Cui et al. (2019) introduced a re-weighting scheme based on the effective number of samples per class to achieve a class-balanced loss. Cao et al. (2019) proposed a label-distribution-aware margin (LDAM) loss to minimize a margin-based generalization bound, which improved generalization on less frequent classes.

Despite their simplicity, re-sampling and re-weighting techniques have limitations in the context of imbalanced regression. First, they fail to fully account for the density of neighboring target values, a critical factor in determining the representativeness of a data point. Yang (2021) emphasized the significance of neighborhood density in imbalance regression. Specifically, a low-frequency point within a dense neighborhood may be adequately represented, while one in a sparse neighborhood remains underrepresented. Secondly, linear interpolation techniques like SMOTE can be ineffective and may degrade performance when generating synthetic samples for high-dimensional data, a common scenario with modern large pre-trained models. Third, the absence of distinct class boundaries in regression tasks poses challenges for the direct application of these methods to regression scenarios.

These limitations highlighted the need for innovative solutions to learn robust representations in imbalanced regression tasks, moving beyond traditional re-sampling or re-weighting techniques.

**Deep Imbalance Regression**:

Deep Imbalanced Regression (DIR), a concept introduced by Yang et al. (2021), addresses the inherent imbalance that are often found in real-world regression tasks. Such challenges of imbalanced data is more intense in deep learning models due to their tendency to produce overconfident predictions that may further amplify the impact of skewed distributions. The goal of DIR is to learn robust representations from imbalanced and skewed data, ensuring that these representations generalize effectively across the entire spectrum of target values.

Yang et al. (2021) proposed feature distribution smoothing (FDS), a technique that smooths feature distributions by transferring statistics between neighboring target bins. This aims to correct potentially biased feature distribution estimates, particularly for underrepresented targets. Based on this insight, recent research has explored achieving this alignment through specialized loss functions. Gong et al. (2022) introduced RankSim, incorporating a ranking loss as a regularizer to effectively capture both local and distant relationships. Zha et al. (2024) proposed Rank-N-Contrast (RNC), which models the ranking loss within a contrastive learning framework to tackle data imbalance. Notably, Rank-N-Contrast has achieved state-of-the-art performance on the Deep Imbalanced Regression (DIR) benchmark established byYang et al. (2021).

In RNC, samples are ranked according to their target distances, and then contrasted against each other based on their relative rankings. Each data sample is sequentially assigned as an anchor point. The distance between this anchor point and every other data sample within the batch is calculated. Based on these distances, data samples are grouped into positive pairs (similar to the anchor) or negative pairs (dissimilar to the anchor). Given an anchor $i$, the similarity in feature space of any other data sample $j$ is measured using the cosine similarity $sim(v_i, v_j)$ where $v_i, v_j$ denote the feature vectors of sample $i$ and $j$, respectively. The set $S_{i,j} := \{k | k \neq i, d(i,k) \geq d(i,j)\}$ denotes the set of samples with larger label distance than $j$ w.r.t. $i$, where $d(i,j)$ is the label distance between two samples $i, j$. The per-sample RNC loss is defined as:

$$\mathcal{L}_{RNC}^{(i)} = -\frac{1}{N-1} \sum_{j \neq i} \log \frac{\exp(sim(v_i, v_j)/\tau)}{\sum_{k \in S_{i,j}} \exp(sim(v_i, v_k)/\tau)}$$

Despite the success of Rank-N-Contrast, it had a significant limitation: it did not consider varying label distances in negative samples, disregarding the impact of negative samples further from the

anchor in the label space, which should ideally provided a stronger contrastive signal than closer ones. Figure 2 illustrated this issue. The top image presented positive and negative pairs within a batch containing posts with popularity scores $\{1, 3, 4, 8\}$. The bottom image showed another batch containing scores $\{1, 3, 4, 15\}$. In this scenario, for the positive pair $\{3, 4\}$, the top batch had negative samples $\{3, 1\}$ and $\{3, 8\}$, and the bottom batch had negative samples $\{3, 1\}$ and $\{3, 15\}$. Similarly, for the positive pair $\{3, 1\}$, the top batch had one negative sample $\{3,8\}$ the bottom batch had negative sample $\{3, 15\}$. Under Rank-N-Contrast, both negative samples $\{3, 8\}$ and $\{3, 15\}$ contributed equally to the overall loss, overlooking the impact of the more popular post with score 15.

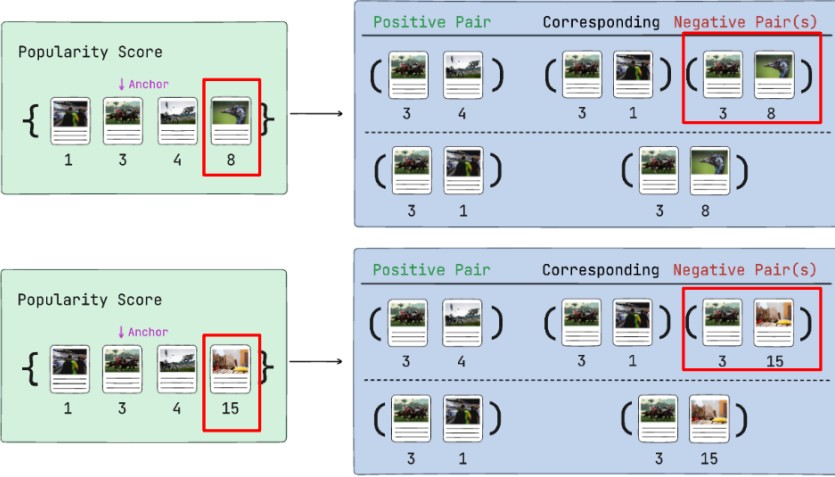

Figure 2: RNC loss treats negative pairs $\{3, 8\}$ and $\{3, 15\}$ equally in both batches, neglecting the impact of the larger label distance posed by the higher popularity score of 15.

### 2.3 OUR CONTRIBUTION

In this paper we refined Rank-N-Contrast (Zha et al., 2024) to overcome its limitation of not distinguishing between negative samples based on their label distances. We introduce a weighting mechanism that incorporates label distance information into the contrastive regression loss. Experimental results demonstrated that our approach fostered a more uniform feature space and significantly improved robustness on extremely rare and even unseen labels. As for our framework, we followed the multi-modal feature extraction framework proposed by Kim et al. (2020) as illustrated in Figure 1 for its simplicity.

## 3 METHODOLOGY

### 3.1 PROBLEM DEFINITION

Given a new post $v$ by user $u$, our objective is to predict its popularity $s$, defined as the expected number of attentions it would received if published at time $t$ on social media. Popularity can be quantified using various dynamic indicators (e.g., views, likes, clicks) across different social media platforms. In our dataset, the "view count" serves as a fundamental indicator of post popularity. To mitigate the wide variations in view counts among photos (ranging from zero to millions), a log-normalization function is applied:

$$s = \log_2 \frac{r}{d} + 1 \tag{1}$$

where $s$ is the normalized popularity, $r$ is the view count, and $d$ is the number of days since posting.

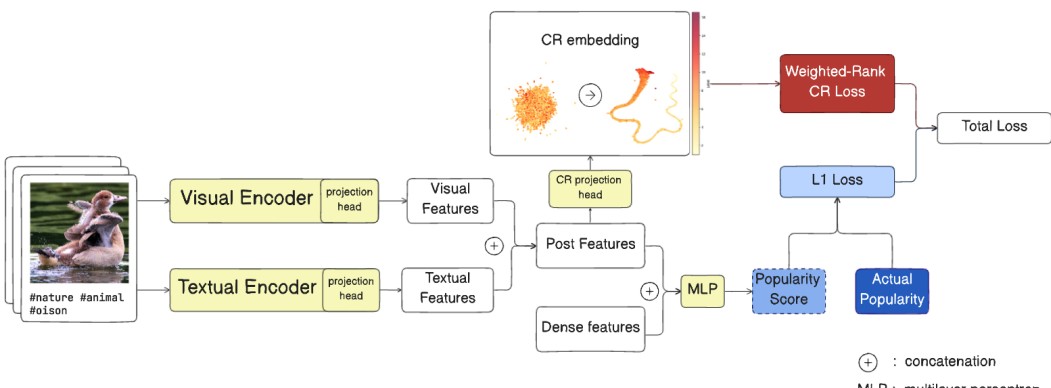

Figure 3: Overview of our proposed framework

## 3.2 PROPOSED FRAMEWORK

We leveraged pre-trained visual and textual models as feature encoders to extract multi-modal features. These features were then concatenated with additional dense features to create a comprehensive input for downstream prediction. The concatenated features were fed into a Multi-Layer Perceptron (MLP) to predict the popularity score. We also incorporated our proposed Weighted-Rank CR loss as a regularizer and calculated the contrastive regression loss alongside the L1 loss, these two losses were combined in a multi-task learning approach, with equal weighting assigned to each loss. This joint optimization process encouraged the feature encoders to learn more robust representations while simultaneously improving the prediction objective during training. Figure 3 illustrates an overview of our framework.

## 3.3 POST REPRESENTATION EXTRACTION

Following the approach of (Kim et al., 2020), we utilized pre-trained models to extract features from both the visual and textual components of the posts. For the visual features, the image preprocessing involved the following steps: (1) conversion to RGB color space, (2) resizing to a 224x224 pixel resolution, (3) subsequent normalization. After preprocessing, we employed the Vision Transformer (VIT) (Dosovitskiy et al., 2021)) to extract the visual features $f_i$. As for textual features, we utilized the hashtags within the social media posts, represented as a list of keywords. By concatenating these keywords, we then leveraged the Sentence Transformer (Reimers and Gurevych, 2019) to extract the textual features $f_t$. Finally, we concatenated $f_i$ and $f_t$ to obtain the comprehensive post features $f_p$.

## 3.4 DENSE FEATURES

Besides the visual and textual inputs, we also used the following dense features provided by the dataset: *userIsPro*: whether the user belong to pro member. *postCount*: The number of posted photo by the user. *photoFirstDateTaken*: The date of the first photo taken by the user. *postDate*: the publish timestamp of the post.

## 3.5 WEIGHTED-RANK CR

We proposed Weighted-Rank CR loss that contrasts negative samples based on their relative label distance with respect to anchor. Following the notation in Rank-N-Contrast (Zha et al., 2024), for an anchor vector $v_i$ and another sample $v_j$ in the batch, we define $S_{i,j}$ as the set of samples whose label distance from $v_i$ are greater than that of $v_j$. In our Weighted-Rank CR loss, we incorporate a weighting mechanism for negative sample pairs such that their contrastive signal is weighted by the relative label distance with respect to anchor. The weight for a negative pair $\{v_i, v_k\}$ is denoted as $w_{i,k}$. We simplified $\exp(\text{sim}(v_i, v_j)/\tau)$ to $e_\tau(v_i, v_j)$, where $sim$ denotes the cosine similarity, and $\tau$ is the temperature hyperparameter in contrastive learning that controls the sensitivity of the

relationship between embedding similarity and the contrastive loss. The per sample Weighted-Rank CR loss can be defined as:

$$\frac{1}{N-1} \sum_{j=1, j \neq i}^{N} -\log \left( \frac{e_\tau(v_i, v_j)}{\sum_{v_k \in S_{i,j}} w_{ik} \cdot e_\tau(v_i, v_k)} \right) \tag{2}$$

$$e_\tau(v_i, v_j) = \exp(sim(v_i, v_j)/\tau) \tag{3}$$

To validate the effectiveness of our weighting mechanism, we conducted experiments on a curated dataset derived from the SMPD (see Section 4) with a skewed distribution for the training phase, and a balanced, uniform distribution for the testing phase (Zha et al., 2024). We evaluated various weighting strategies including logarithmic, linear, quadratic, and exponential weighting on the uniform distributed test set. The results, presented in Table 1, support our hypothesis that a stronger emphasis on contrastive signals based on label distance leads to improved performance. Notably, the exponential weighting strategy, represented by $(1 + \alpha)^d$, where $d$ is the label distance, achieved the best performance. The quadratic weighting strategy, $d^2 + 1$, is closely behind. In contrast, linear weighting $(d + 1)$ and logarithmic weighting $\log(d + 1) + 1$ did not outperform the baseline Rank-N-Contrast method. These findings reinforced our hypothesis that prioritizing distant negative samples in the contrastive loss can enhance the effectiveness of contrastive regression.

Table 1: Performance metrics of different weighting strategies.

| Weighting Strategy | metrics | |
|---|---|---|
| | MAE | SRC |
| RNC (baseline) | 2.198 | 0.838 |
| $\log(d + 1) + 1$ | 2.715 | 0.510 |
| $d + 1$ | 2.642 | 0.579 |
| $d^2 + 1$ | 2.175 | 0.838 |
| $(1 + \alpha)^d$ | **2.142** | **0.841** |

As a result, we incorporated an exponential weighting on label distance in our proposed Weighted-Rank CR loss to amplify the feature space distance for more distant negative pairs. Let $w_{i,k}$ denote the weight assigned to the negative sample pair $\{i, k\}$ and $d$ denote the absolute label difference between sample $i$ and $k$. Then, $w_{i,k}$ is calculated as in (4), where $\alpha$ is a hyperparamter that controls the slope of $w_{i,k}$. In our experiment, we chose $\alpha = 0.4$ so that $w_{i,k}$ is bounded within the range of our label value.

$$w_{i,k} = (1 + \alpha)^{d(i,k)} \tag{4}$$

For example in Figure 2, with Weighted-Rank CR loss, the negative pairs $\{3, 15\}$ and pair $\{3, 8\}$ will now be assigned weights of $(1 + \alpha)^{|3-15|} = 1.4^{12}$ and $(1 + \alpha)^{|3-8|} = 1.4^5$, respectively. Consequently, the post with the higher popularity score of 15 is mapped farther away from the anchor post 3 in the feature space under this weighted scenario. This weighting scheme ensures that negative samples with larger label distances from the anchor have a stronger influence on the contrastive loss, leading to more effective learning of feature representations, especially for rare and extreme labels.

We used a CR projection head to perform contrastive learning on the extracted post features $f_p$. After feature extraction, $f_p$ were passed through the CR projection head. Here we denoted the output of CR projection head as $f_p^{cr}$. The Weighted-Rank CR loss was then computed on $f_p^{cr}$, enforcing the feature encoders to align the feature space with the corresponding label distances. In parallel, $f_p$ was also fed into a Multi-Layer Perceptron (MLP) to generate a predicted popularity score. We calculated the L1 loss between this predicted score and the actual popularity score. Finally, we combined the Weighted-Rank CR loss and the L1 loss in a multi-task learning framework. Both losses were given equal weight, without emphasizing one over the other. This approach ensures

that the model learns robust feature representations while simultaneously optimizing its predictive performance.

# 4 EXPERIMENT SETTING

We utilized the Social Media Prediction Dataset (SMPD) proposed in (Wu et al., 2019), which was collected from Flickr, a major photo-sharing platform. SMPD comprises 486K social multimedia posts from 70K users, and incorporates diverse social media information such as anonymized photo-sharing records, user profiles, web images, text, timestamps, location data, and categories. Table 2 provides a detailed overview of the dataset statistics.

Table 2: Dataset statistics for SMPD.

| Dataset | #Post | #User | #Categories | Temporal Range (Months) | Avg. Title Length | #Customize Tags |
|---------|-------|-------|-------------|-------------------------|-------------------|-----------------|
| SMPD | 486k | 70k | 756 | 16 | 29 | 250k |

We combined the Spearman Ranking Correlation (SRC) and Mean Absolute Error (MAE) to assess model performance. SRC quantifies the ordinal association between predicted and actual popularity rankings, while MAE measures the average prediction error.

SRC is calculated as follows:

$$SRC = \frac{1}{k-1} \sum_{i=1}^{k} \left( \frac{P_i - \bar{P}}{\sigma_P} \right) \left( \frac{\hat{P}i - \tilde{P}}{\sigma_{\hat{P}}} \right) \tag{5}$$

where $k$ is the number of samples, $P_i$ is the actual popularity, $\hat{P}i$ is the predicted popularity, $\bar{P}$ and $\sigma_P$ are the mean and standard deviation of actual popularity, and $\tilde{P}$ and $\sigma_{\hat{P}}$ are the mean and standard deviation of predicted popularity, respectively.

MAE is calculated as follows:

$$\text{MAE} = \frac{1}{k} \sum_{i=1}^{n} \left| \hat{P}_i - P_i \right| \tag{6}$$

The goal of SMPP is to enhance both ranking accuracy and prediction accuracy by minimizing the MAE and maximizing the SRC.

The model architecture and hyperparameters are detailed in the Appendix.

# 5 EVALUATION RESULTS

## 5.1 EXPERIMENT ON SOCIAL MEDIA PREDICTION DATASET (SMPD)

To evaluate our proposed framework, we utilized the test API provided by the SMP Challenge (Wu et al., 2019). This API allows us to upload our prediction results and obtain the corresponding performance metrics through an online interface. Our experiments included three different modalities: text only, image only, and multi-modal inputs. The evaluation results are presented in Table 3. The numbers in parentheses represent the relative differences compared to the Vanilla baseline. Green values indicate a decrease in Mean Absolute Error (MAE) or an increase in Spearman Rank Correlation (SRC), signifying an improvement. Conversely, red values indicate a decline in performance. As can be seen, Weighted-Rank CR outperforms both the vanilla approach (direct L1 loss fitting) and Rank-N-Contrast in terms of MAE and SRC across all three modalities: Tags, Image, and Tags + Image. While Rank-N-Contrast shows improvements in MAE and SRC for the Tag-only and Image-only settings, its performance deteriorates with higher MAE when considering the Tags + Image modality. This decline can be attributed to the inherent complexity of multi-modal data, where integrating text and image information demands a more sophisticated approach to capture

effective representations across different modalities. Our Weighted-Rank CR loss, by addressing data imbalance, is better equipped to handle the challenges presented by the Tags + Image input and consequently, generates more generalized representations.

Table 3: Performance metrics for different training objectives and input types.

| Input | Vanilla (L1) | | Rank-N-Contrast | | Weighted-Rank CR | |
|---|---|---|---|---|---|---|
| | MAE | SRC | MAE↓ | SRC↑ | MAE↓ | SRC↑ |
| Tags | 2.040 | 0.468 | 1.995 (+0.045) | 0.483 (+0.015) | **1.925** (+0.115) | **0.499** (+0.031) |
| Image | 2.262 | 0.301 | 2.214 (+0.048) | 0.303 (+0.002) | **2.183** (+0.079) | **0.310** (+0.009) |
| Tags + Image | 1.955 | 0.473 | 2.001 (-0.045) | 0.501 (+0.028) | **1.901** (+0.054) | **0.504** (+0.031) |

## 5.2 EXPERIMENT ON CURATED DATASETS

We curated two datasets with more imbalance distribution to test the robustness of Weighted-Rank CR. First, we sampled a subset from SMPD training dataset with only few data points at both ends. Figure 4 illustrates the distribution of this sampled dataset.

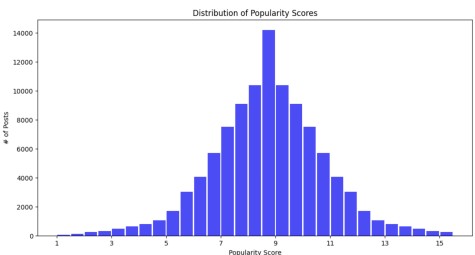

Figure 4: The distribution of the sampled dataset, with very few data points on both ends.

We visualized the MAE improvement across different label bins in Figure 5. The x-axis represents the label ranges, with the top portion of the figure depicting the data distribution (y-axis showing the number of posts), and the bottom portion displaying the MAE improvement (y-axis indicating the MAE difference). Positive values (in green) signify a lower MAE for that label bin, while negative values (in red) signify a higher MAE. The results demonstrate that contrastive regression substantially reduces the MAE for rarely seen data points, particularly at both extremes of the distribution. Furthermore, we visualized the MAE improvement of Weighted-Rank CR over Rank-N-Contrast in Figure 6. The results demonstrate that Weighted-Rank CR surpasses Rank-N-Contrast in terms of MAE for the less frequent label bins within the skewed-sampled dataset.

We also curated another more imbalanced dataset by removing data points with popularity scores below 4.0 and above 13.0. Figure 7 illustrated the distribution of this dataset. The MAE improvement across different label bins for this dataset is illustrated in Figure 8. Figure 9 visually represents the MAE improvement of Weighted-Rank CR compared to Rank-N-Contrast on this more imbalanced dataset. The results again demonstrate that Weighted-Rank CR consistently achieves lower MAE than Rank-N-Contrast on most label bins, even in this more challenging scenario.

## 6 CONCLUSION

In this paper, we delved into the challenges of imbalanced regression in social media popularity prediction, highlighting the limitations of existing contrastive learning methods like Rank-N-Contrast. We proposed Weighted-Rank CR loss, a contrastive learning loss that incorporates label distance information into the Rank-N-contrast loss function, thereby enhancing the model's ability to learn effective representations for rare and extreme labels.

Our experiments on the Social Media Prediction Dataset (SMPD) showed that Weighted-Rank CR outperforms the baseline methods (including the current state-of-the-art contrastive regression ap-

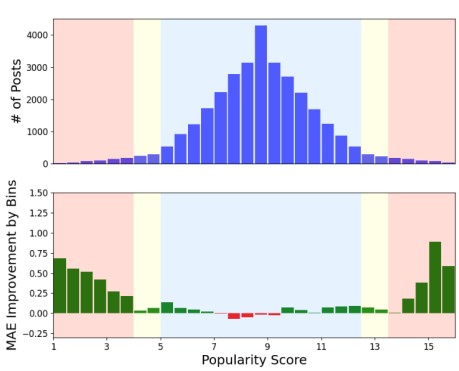 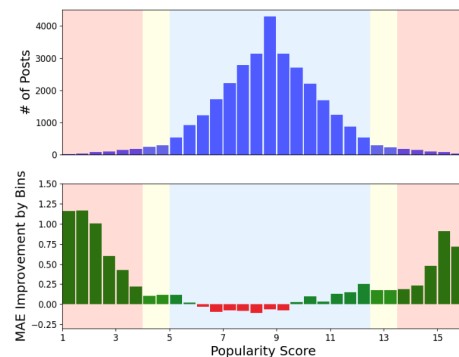

(a) The MAE improvement of Rank-N-Contrast over the Vanilla approach.

(b) The MAE improvement of Weighted-Rank-CR over the Vanilla approach.

Figure 5: The MAE improvement of both Rank-N-Contrast and Weighted-Rank-CR compared to the Vanilla approach. Positive values (in green) signify a lower MAE on the label bin, and negative values (in red) signify a higher MAE on the label bin.

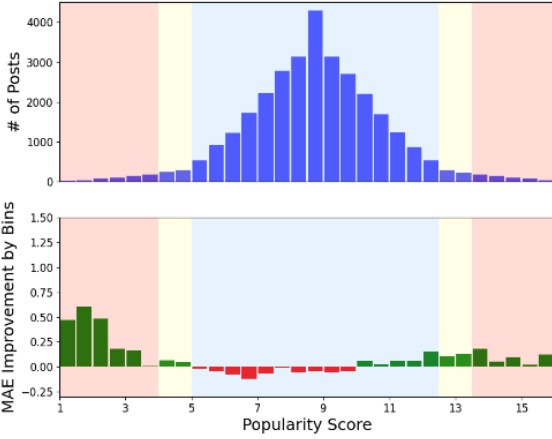

Figure 6: The MAE improvement of Weighted-Rank-CR over Rank-N-Contrast.

proach Rank-N-Contrast) in both ranking and prediction accuracy. Our approach is particularly effective in handling imbalanced datasets, where rare labels are often underrepresented. In conclusion, our research contributes to the growing body of work addressing the challenges of imbalanced learning in Social Media Popularity Prediction (SMPP). The proposed Weighted-Rank CR method offers a promising avenue for future research, with potential applications in various domains where data imbalance poses a significant challenge.

Future work may explore more sophisticated weighting mechanisms could potentially lead to further performance improvements in contrastive regression. Additionally, conducting experiments on a wider range of datasets and downstream tasks would help validate the effectiveness of Weighted-Rank CR in various settings.

## REFERENCES

Paula Branco, Luís Torgo, and Rita P Ribeiro. 2017. SMOGN: a pre-processing approach for imbalanced regression. In *First international workshop on learning with imbalanced domains: Theory*

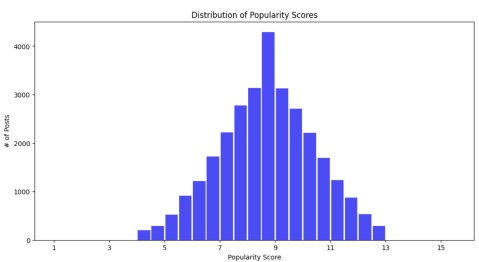

Figure 7: The distribution of the sampled dataset, with no data points on both ends.

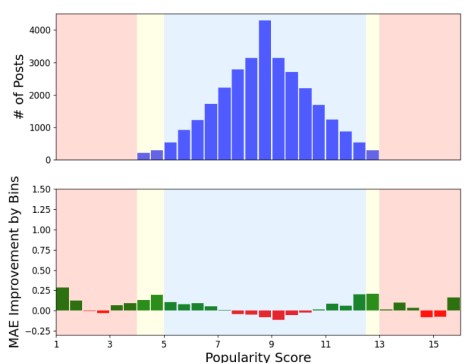

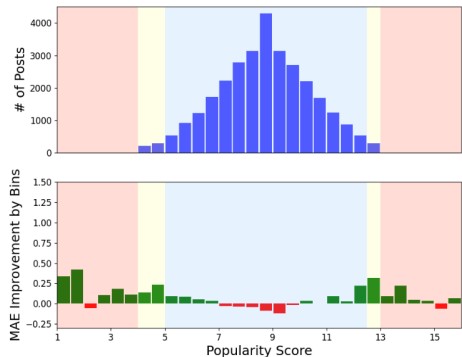

(a) The MAE improvement of Rank-N-Contrast over the Vanilla approach.

(b) The MAE improvement of Weighted-Rank-CR over the Vanilla approach.

Figure 8: The MAE improvement of both Rank-N-Contrast and Weighted-Rank-CR compared to the Vanilla approach on a dataset that data points at both extremes are removed.

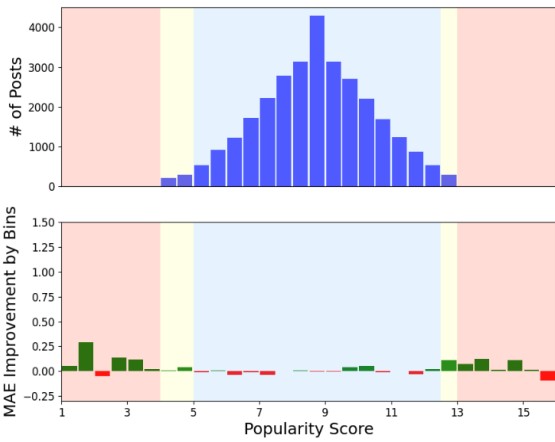

Figure 9: The MAE improvement of Weighted-Rank-CR over Rank-N-Contrast on a dataset where data points at both extremes are removed.

*and applications*. PMLR, 36–50.

Kaidi Cao, Colin Wei, Adrien Gaidon, Nikos Arechiga, and Tengyu Ma. 2019. Learning imbalanced datasets with label-distribution-aware margin loss. *Advances in neural information processing systems* 32 (2019).

Nitesh V Chawla, Kevin W Bowyer, Lawrence O Hall, and W Philip Kegelmeyer. 2002. SMOTE: synthetic minority over-sampling technique. *Journal of artificial intelligence research* 16 (2002), 321–357.

Yin Cui, Menglin Jia, Tsung-Yi Lin, Yang Song, and Serge Belongie. 2019. Class-balanced loss based on effective number of samples. In *Proceedings of the IEEE/CVF conference on computer vision and pattern recognition*. 9268–9277.

Keyan Ding, Ronggang Wang, and Shiqi Wang. 2019. Social media popularity prediction: A multiple feature fusion approach with deep neural networks. In *Proceedings of the 27th ACM International Conference on Multimedia*. 2682–2686.

Alexey Dosovitskiy, Lucas Beyer, Alexander Kolesnikov, Dirk Weissenborn, Xiaohua Zhai, Thomas Unterthiner, Mostafa Dehghani, Matthias Minderer, Georg Heigold, Sylvain Gelly, Jakob Uszkoreit, and Neil Houlsby. 2021. An Image is Worth 16x16 Words: Transformers for Image Recognition at Scale. In *International Conference on Learning Representations*.

Francesco Gelli, Tiberio Uricchio, Marco Bertini, Alberto Del Bimbo, and Shih-Fu Chang. 2015. Image popularity prediction in social media using sentiment and context features. In *Proceedings of the 23rd ACM international conference on Multimedia*. 907–910.

Yu Gong, Greg Mori, and Frederick Tung. 2022. RankSim: Ranking Similarity Regularization for Deep Imbalanced Regression. In *International Conference on Machine Learning (ICML)*.

Haibo He and Yunqian Ma. 2013. Imbalanced learning: foundations, algorithms, and applications. (2013).

Xin Jin, Andrew Gallagher, Liangliang Cao, Jiebo Luo, and Jiawei Han. 2010. The wisdom of social multimedia: using flickr for prediction and forecast. In *Proceedings of the 18th ACM international conference on Multimedia*. 1235–1244.

Seungbae Kim, Jyun-Yu Jiang, Masaki Nakada, Jinyoung Han, and Wei Wang. 2020. Multimodal post attentive profiling for influencer marketing. In *Proceedings of The Web Conference 2020*. 2878–2884.

Xin Lai, Yihong Zhang, and Wei Zhang. 2020. Hyfea: winning solution to social media popularity prediction for multimedia grand challenge 2020. In *Proceedings of the 28th ACM International Conference on Multimedia*. 4565–4569.

Philip J McParlane, Yashar Moshfeghi, and Joemon M Jose. 2014. ” Nobody comes here anymore, it's too crowded”; Predicting Image Popularity on Flickr. In *Proceedings of international conference on multimedia retrieval*. 385–391.

Nils Reimers and Iryna Gurevych. 2019. Sentence-BERT: Sentence Embeddings using Siamese BERT-Networks. In *EMNLP/IJCNLP (1)*, Kentaro Inui, Jing Jiang, Vincent Ng, and Xiaojun Wan (Eds.). Association for Computational Linguistics, 3980–3990.

Luís Torgo, Paula Branco, Rita P Ribeiro, and Bernhard Pfahringer. 2015. Resampling strategies for regression. *Expert systems* 32, 3 (2015), 465–476.

Luís Torgo, Rita P Ribeiro, Bernhard Pfahringer, and Paula Branco. 2013. Smote for regression. In *Portuguese conference on artificial intelligence*. Springer, 378–389.

Bo Wu, Wen-Huang Cheng, Peiye Liu, Bei Liu, Zhaoyang Zeng, and Jiebo Luo. 2019. SMP Challenge: An Overview of Social Media Prediction Challenge 2019. In *Proceedings of the 27th ACM International Conference on Multimedia*.

Jianmin Wu, Liming Zhao, Dangwei Li, Chen-Wei Xie, Siyang Sun, and Yun Zheng. 2022. Deeply exploit visual and language information for social media popularity prediction. In *Proceedings of the 30th ACM International Conference on Multimedia*. 7045–7049.

Kele Xu, Zhimin Lin, Jianqiao Zhao, Peicang Shi, Wei Deng, and Huaimin Wang. 2020. Multimodal deep learning for social media popularity prediction with attention mechanism. In *Proceedings of the 28th ACM International Conference on Multimedia*. 4580–4584.

Yuzhe Yang. 2021. *Strategies and Tactics for Regression on Im-balanced Data*. https://towardsdatascience.com/strategies-and-tactics-for-regression-on-imbalanced-data-61eeb0921fca accessed June 22, 2024.

Yuzhe Yang, Kaiwen Zha, Yingcong Chen, Hao Wang, and Dina Katabi. 2021. Delving into deep imbalanced regression. In *International conference on machine learning*. PMLR, 11842–11851.

S Yen and Y Lee. 2006. Under-sampling approaches for improving prediction of the minority class in an imbalanced dataset. *Lecture notes in control and information sciences* 344 (2006), 731.

Kaiwen Zha, Peng Cao, Jeany Son, Yuzhe Yang, and Dina Katabi. 2024. Rank-n-contrast: learning continuous representations for regression. *Advances in Neural Information Processing Systems* 36 (2024).

# A   APPENDIX

Table 4 outlines the training configuration including hardware specifications and hyperparameters. We fixed these settings in the main experiments discussed in Section 5. Specifically, we chose the largest batch size we can afford under the hardware limitation, and $\tau$ represents the temperature which controls the sensitivity of the feature similarity during contrastive learning.

Table 4: Training configuration.

| hardware | RTX 4080 |
|---|---|
| **number of epochs** | 10 |
| **learning rate** | 3e-4 |
| **random seed** | 3407 |
| **batch size** | 128 |
| $\tau$ | 0.05 |

The model architecture of our framework is as below:

- **Backbone Model**: We used a pre-trained VIT model for visual encoder, and a pre-trained sentence-transformers for textual encoder.

- **Encoder Projection Head**: Both the visual and textual projection heads adhere to the architecture outlined in Figure 10a. The input feature tensors, initially of dimension 384, are first expanded to 1536 dimensions and then subjected to a non-linear transformation using LeakyReLU activation. Finally, a linear layer projects the output tensor to 128 dimensions.

- **CR Projection Head**: As illustrated in Figure 10b, the input size of 256 represents the concatenated visual and textual features. We then apply a ReLU non-linear transformation, and finally a linear layer to reduce the output tensor to 64 dimensions.

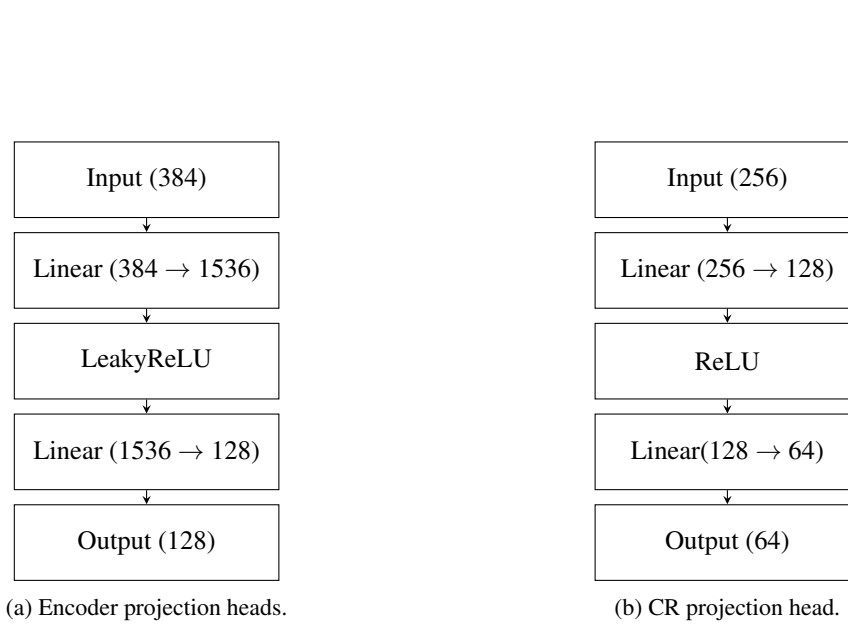

(a) Encoder projection heads.                    (b) CR projection head.

Figure 10: Projection heads.

