# OpenReview forum: "Weighted-Rank Contrastive Regression for Robust Learning on Imbalance Social Media Popularity Prediction"
_ICLR.cc/2025/Conference — Submitted to ICLR 2025_

### Official Review · Reviewer_KD8Z · 2024-10-27

**Soundness:** 2
**Presentation:** 1
**Contribution:** 2
**Rating:** 3
**Confidence:** 4

**Summary:**

This paper investigates the challenges associated with Social Media Popularity Prediction (SMPP), particularly focusing on the issue of data imbalance. The authors introduce a novel Weighted-Rank Contrastive Regression loss (Weighted-Rank CR loss) designed to enhance the model's predictive power by assigning weights to negative samples. The study employs pre-trained visual and textual models to extract multimodal features, which are then combined with additional dense features for downstream prediction tasks. By integrating Weighted-Rank CR loss with L1 loss within a multi-task learning framework, the authors effectively demonstrate the method's efficacy in predicting social media popularity.

**Strengths:**

1. The paper introduces a Weighted-Rank Contrastive Regression loss to address data imbalance, highlighting the importance of negative samples that are further from anchor points through higher weights (i.e., different negative samples contribute differently to the model's popularity predictions).
2. The methodology presented is easy to understand.
3. The experimental results appear promising.

**Weaknesses:**

1. The novelty of this paper is limited. It seems to only add different weights to different negative samples in contrastive loss to distinguish their contributions to find out the hard negative samples. Compared to the previous work directed at learning the features of negative samples using contrast (Kalantidis Y, Sariyildiz M B, Pion N, et al. Hard negative mixing for contrastive learning[J]. Advances in neural information processing systems, 2020, 33: 21798-21809.), the paper is of limited innovation. The article seems to be just a small improvement or trick for contrastive loss in the context of unbalanced regression, where hard negative samples in contrastive learning are going to improve the final result unquestionably. The authors' approach is viewed as explicitly highlighting hard negative samples using a simple weighting method in the paper formulation, and the authors in the article do not give the way of selecting the positive and negative samples.
2. The writing quality is poor. For example, the model diagram in Figure 3 does not significantly illustrate the contributions of the proposed method, and the format of Table 2 is not standardized; Figures 7, 8, and 9 appear in the references section, with some references formatted incorrectly.
3. The validation dataset is limited. The paper only uses one dataset for validation, which raises concerns about the model's generalizability and practical applicability. Some additional commonly used datasets in the field of SMPP can be added, such as TPIC17 (Wu, B., Cheng, W.-H., Zhang, Y., Huang, Q., Li, J., Mei, T., 0000. Sequential prediction of social media popularity with deep temporal context networks, arXiv preprint arXiv:1712.04443.)
4. This paper compares fewer methods and does not adequately compare the performance with other state-of-the-art methods (e.g., Yang Y, Zha K, Chen Y, et al. Delving into deep imbalanced regression[C]//International conference on machine learning. PMLR, 2021: 11842-11851). Meanwhile, the lack of comparative analyses in the experimental part may reduce the persuasiveness of the results.

**Questions:**

1. The novelty of this paper is limited. It seems to only add different weights to different negative samples in contrastive loss to distinguish their contributions to find out the hard negative samples. Compared to the previous work directed at learning the features of negative samples using contrast (Kalantidis Y, Sariyildiz M B, Pion N, et al. Hard negative mixing for contrastive learning[J]. Advances in neural information processing systems, 2020, 33: 21798-21809.), the paper is of limited innovation. The article seems to be just a small improvement or trick for contrastive loss in the context of unbalanced regression, where hard negative samples in contrastive learning are going to improve the final result unquestionably. The authors' approach is viewed as explicitly highlighting hard negative samples using a simple weighting method in the paper formulation, and the authors in the article do not give the way of selecting the positive and negative samples.
2. The validation dataset is limited. The paper only uses one dataset for validation, which raises concerns about the model's generalizability and practical applicability. Some additional commonly used datasets in the field of SMPP can be added, such as TPIC17 (Wu, B., Cheng, W.-H., Zhang, Y., Huang, Q., Li, J., Mei, T., 0000. Sequential prediction of social media popularity with deep temporal context networks, arXiv preprint arXiv:1712.04443.)
3. This paper compares fewer methods and does not adequately compare the performance with other state-of-the-art methods (e.g., Yang Y, Zha K, Chen Y, et al. Delving into deep imbalanced regression[C]//International conference on machine learning. PMLR, 2021: 11842-11851). Meanwhile, the lack of comparative analyses in the experimental part may reduce the persuasiveness of the results.

---

### Official Review · Reviewer_XBnU · 2024-10-30

**Soundness:** 2
**Presentation:** 2
**Contribution:** 2
**Rating:** 3
**Confidence:** 4

**Summary:**

This paper tackles the challenge of imbalanced regression in social media popularity prediction (SMPP) by proposing an enhanced approach. The authors introduce Weighted-Rank CR loss, an extension of the existing Rank-N-Contrast method (Zha et al., 2024), which integrates label distance information into the contrastive regression framework. This enhancement strengthens the model’s ability to manage rare and extreme labels, improving robustness with imbalanced data. Experimental results show that Weighted-Rank CR loss significantly outperforms baseline methods, including the current state-of-the-art Rank-N-Contrast, on the Social Media Prediction Dataset (SMPD). A key contribution of this work is the novel weighting mechanism in contrastive learning loss, along with the effective integration of multi-modal inputs (text and image) for more accurate popularity prediction.

**Strengths:**

- Originality: This paper builds on Rank-N-Contrast (Zha et al., 2024) by introducing a novel weighting mechanism that incorporates label distance into the contrastive regression framework, addressing limitations in handling imbalanced regression for social media popularity prediction (SMPP). The approach’s adaptability suggests potential applications across other domains facing similar challenges.

- Quality: The authors present a rigorous experimental design, evaluating their method on the Social Media Prediction Dataset (SMPD) and comparing it to baseline models, including Rank-N-Contrast. Results show notable improvements in prediction accuracy and robustness, though testing on additional datasets could further validate these findings.

- Significance: This work contributes to both social media analysis and imbalanced regression, with Weighted-Rank CR loss showing promise for broader application to other continuous regression tasks with imbalanced data, though additional validation is needed.

**Weaknesses:**

- Limited Dataset Evaluation: The experiments are confined to the Social Media Prediction Dataset (SMPD). Testing on additional datasets would strengthen claims of generalizability.
- Lack of Theoretical Justification for Weighting Mechanism: While the Weighted-Rank CR loss is a central contribution, the rationale behind its specific weighting mechanism (Table 1) could be expanded with a more thorough theoretical explanation.
- Narrow Baseline Comparison: The comparison primarily focuses on Rank-N-Contrast. Including other popular imbalanced learning methods or deep regression models would provide a more comprehensive evaluation.
- Presentation Quality Issues: The paper’s presentation could be improved; figures are not vector graphics, and some tables appear poorly formatted.

**Questions:**

- Have you considered further discussing the potential of applying this method to other domains? For example, could this method be applied to tasks in healthcare, finance, or other regression problems?
- Is there a way to avoid the performance degradation observed in Figure 5, particularly in regions with a larger amount of data, due to the effects of Weighted-Rank CR?

---

### Official Review · Reviewer_zoGs · 2024-10-31

**Soundness:** 1
**Presentation:** 1
**Contribution:** 1
**Rating:** 1
**Confidence:** 2

**Summary:**

The paper introduces a Weighted-Rank CR loss for Social Media Popularity Prediction (SMPP), addressing data imbalance by leveraging contrastive learning. This approach outperforms both the vanilla method and the state-of-the-art Rank-N-Contrast, effectively improving engagement forecasting.

**Strengths:**

1.	The paper is clearly presented.
2.	The authors summarize other related works clearly.

**Weaknesses:**

1.	The novelty is limited. The contribution is adding a weight into the existing contrastive regression loss.
2.	Experiment is weak. Lack of significant comparison methods related to the task: social media popularity prediction, which the authors are dealing with.
[1]Z. Cheng, J. Zhang, X. Xu, G. Trajcevski, T. Zhong, F. Zhou, “Retrieval-Augmented Hypergraph for Multimodal Social Media Popularity Prediction.” in KDD 2024.
[2]X. Chen, W. Chen, C. Huang, Z. Zhang, L. Duan, Y. Zhang, “Double-Fine-Tuning Multi-Objective Vision-and-Language Transformer for Social Media Popularity Prediction.” in ACM Multimedia 2023.

**Questions:**

1.	The authors mentioned that Rank-N-Contrast is published in 2024 (page 1, line 21, Rank-N-Contrast (Zha et al., 2024)), however, according to OpenReview (https://openreview.net/forum?id=WHedsAeatp) and NeurIPS Proceedings Search (https://proceedings.neurips.cc/paper_files/paper/2023/hash/39e9c5913c970e3e49c2df629daff636-Abstract-Conference.html), Rank-N-Contrast is published on NeurIPS 2023.
2.	The authors’ organization of the article is terrible. Most of the space of the article is spent on other methods, while the proposed method’s introduction focuses on only one weight coefficient (Equation 4 and Table 1). At the same time, there is a big difference between Table 1’s result and Table 3’s result, it is difficult to convince that the weight coefficient (Equation 4) plays a positive effect in the whole framework. The authors should add ablation studies to convince the positive effect of the weight coefficient (Equation 4).
3.	The authors’ contributions seem to be too little. They used only one dataset, compared with only one method, without any other SOTAs related to social media popularity prediction. At the same time, I doubt the generalization of the Weighting Strategy (Table 1), is the coefficient effective and universal on other methods and other datasets?
4.	The design of the proposed weights seem not relative to the specific task. So why evaluate the method on the task of Social Media Prediction? Why not using other datasets such as those utilized in Rank-N-Contrast paper?
5.	There are some minor typos:
a)	Page 1, line 32, “decisions” should be “decisions.”.
b)	Page 3, line 112, “lables” should be “labels”.
c)	Table 2 has no borders.

---

### Official Review · Reviewer_ZTVy · 2024-11-03

**Soundness:** 3
**Presentation:** 3
**Contribution:** 3
**Rating:** 5
**Confidence:** 4

**Summary:**

The paper proposes a novel method for Social Media Popularity Prediction (SMPP) using a technique called Weighted-Rank Contrastive Regression , aiming to address data imbalance issues in popularity prediction tasks. Building on the Rank-N-Contrast method, the authors introduce a weighting mechanism that considers the label distance between samples, enhancing the contrastive learning process by differentiating the importance of negative pairs based on popularity differences. Extensive experiments conducted on the Social Media Prediction Dataset (SMPD) and selected datasets demonstrate that this approach outperforms baseline and state-of-the-art models, particularly in scenarios with skewed and imbalanced popularity scores.

**Strengths:**

1. The proposed Weighted-Rank CR method is an innovative extension of Rank-N-Contrast and aligns well with the requirements of the SMPP task.
2. Experimental validation on different feature configurations demonstrates its effectiveness.

**Weaknesses:**

​1. Distance weighting has been developed extensively in the field of machine learning, and this method merely adds distance weighting to an existing approach, which is not novel.
2. The paper uses a large dataset, SMPD, and creates two subsets with more imbalanced distributions through sampling. However, there is only one actual dataset, which may not sufficiently demonstrate the method's effectiveness.
3. The dependence of the method on the weighting factor alpha has not been studied.
4. The paper lacks an analysis of why an exponential weighting strategy was used instead of other weighting mechanisms, or how it interacts with different data characteristics. This omission makes it difficult to assess the model's adaptability to other weighting mechanisms or data distributions.

**Questions:**

1. Does this method perform well on other datasets?
​2. Can this method be extended to other domains of imbalanced data regression tasks? If so, what adjustments would be needed?
​3. Has this weighting mechanism been combined with alternative loss functions outside of Rank-N-Contrast to understand the model's adaptability to other contrastive or regression methods?
​4. How does the computational efficiency of this method compare with that of simpler models?

---

### Meta-Review · Area_Chair_Q4GY · 2024-12-19

**Metareview:**

The submission introduces a method titled "Weighted-Rank Contrastive Regression" aimed at addressing data imbalance in the context of Social Media Popularity Prediction (SMPP). By leveraging contrastive regression with a weighted rank-based approach, the proposed method emphasizes label distance in the learning process, enhancing the differentiation between hard and easy samples. The study demonstrates its approach on the Social Media Prediction Dataset (SMPD), showing improvements over the baseline and the state-of-the-art Rank-N-Contrast method.

While the core idea is straightforward and well-aligned with the problem of imbalance, its overall novelty is limited. The weighting mechanism extends existing concepts in contrastive learning, but the theoretical contribution is minimal, and the implementation appears to be an incremental improvement rather than a groundbreaking advancement. The reliance on a single dataset for validation and the lack of generalization to other tasks or domains further undermine the paper's impact. Additionally, there are unresolved issues related to the interpretability and adaptability of the weighting mechanism, such as why an exponential weighting strategy was chosen over alternatives. Concerns about the scalability of the method, its dependence on hyperparameter tuning (e.g., alpha), and its applicability to other datasets remain unanswered.

The primary strengths of the paper include a clear problem formulation, effective integration of multi-modal inputs, and a systematic comparison against a limited baseline. However, these positives are overshadowed by the weak experimental design, lack of robust evaluations across diverse datasets, insufficient theoretical justification, and presentation issues such as poor formatting and minor typographical errors. Despite some experimental success, the submission fails to make a compelling case for its broader utility and impact, particularly given the limited novelty and evaluation scope.

The decision to reject is based on the following reasons: (1) incremental contribution with limited theoretical innovation, (2) insufficient experimental rigor and dataset diversity, (3) poor justification of methodological choices, and (4) suboptimal presentation quality, which detracts from the paper's clarity and credibility.

**Additional Comments On Reviewer Discussion:**

During the reviewer discussion, the points raised included the paper’s limited novelty (noted by zoGs and XBnU), restricted dataset evaluation (XBnU and KD8Z), and the lack of theoretical justification for the weighting mechanism (raised by multiple reviewers). Concerns about generalization, missing comparisons with state-of-the-art methods, and inconsistencies in experimental results (noted by zoGs) were also highlighted.

The authors attempted to address some of these issues in their rebuttal, primarily by clarifying the rationale behind the weighting mechanism and adding additional insights into their experimental results. However, these responses were insufficient to alleviate the reviewers’ primary concerns. For example, the authors did not provide additional datasets or ablation studies, nor did they adequately address the limited novelty or explain the generalizability of their method.

In weighing these points, I found that the criticisms about novelty, experimental design, and evaluation diversity were well-substantiated and remained unresolved despite the rebuttal. While the method demonstrates potential, it does not meet the high bar for acceptance at this venue due to its incremental nature and limited impact.

---

### Decision · Program_Chairs · 2025-01-22

Reject